# The Effects of National Insurance Coverage Expansion and Genetic Counseling’s Role on *BRCA1/2* Mutation Tests in Breast Cancer Patients

**DOI:** 10.3390/cancers16101865

**Published:** 2024-05-14

**Authors:** Sung Yoon Jang, Youngji Kwak, Joon Young Choi, Dong Seung Shin, Hyunjun Lee, Mina Kim, Boo Yeon Jung, Byung Joo Chae, Jonghan Yu, Jeong Eon Lee, Seok Won Kim, Seok Jin Nam, Jai Min Ryu

**Affiliations:** 1Division of Breast Surgery, Department of Surgery, Jeju National University Hospital, Jeju National University School of Medicine, 15, Aran 13-gil, Jeju-si 63241, Republic of Korea; sandslush@naver.com (S.Y.J.); choejun02@naver.com (J.Y.C.); 2Division of Breast Surgery, Department of Surgery, Chung-Ang University Hospital, Chung-Ang University College of Medicine, 102 Heukseok-ro, Dongjak-gu, Seoul 06973, Republic of Korea; youngji.kwak@cauhs.or.kr; 3Division of Breast Surgery, Department of Surgery, Samsung Medical Center, School of Medicine, Sungkyunkwan University School of Medicine, Seoul 06351, Republic of Korea; ds_lovely@hanmail.com (D.S.S.); bj.chae@samsung.com (B.J.C.); jonghan.yu@samsung.com (J.Y.); jeongeon.lee@samsung.com (J.E.L.); seokwon1.kim@samsung.com (S.W.K.); seokjin.nam@samsung.com (S.J.N.); 4Breast Cancer Center, Samsung Medical Center, Seoul 06351, Republic of Korea; mina216.kim@samsung.com (M.K.); by.jeong@samsung.com (B.Y.J.)

**Keywords:** breast cancer, *BRCA1/2* mutation, Korean national insurance coverage, insurance coverage expansion, genetic counseling, genetic counselor

## Abstract

**Simple Summary:**

This research investigates the effects of South Korea’s national insurance coverage (NIC) expansion and the inclusion of genetic counselors on *BRCA1/2* mutation testing rates in breast cancer patients. By analyzing data from the Samsung Medical Center, the study reveals a notable increase in testing rates following NIC expansion and the addition of genetic counselors. Particularly noteworthy is the rise in testing rates among triple-negative breast cancer (TNBC) patients under 60. Additionally, the involvement of genetic counselors led to a significant increase in follow-up patients undergoing testing. The NIC expansion also broadened insurance coverage for TNBC patients, thereby enhancing testing accessibility. These findings underscore the positive impact of NIC expansion and genetic counselor involvement on improving patient management. Addressing financial barriers to testing and integrating genetic counseling into healthcare practices present promising strategies for advancing early detection and tailored treatment approaches, thereby contributing to global efforts in cancer care and management.

**Abstract:**

Purpose: This study aims to evaluate the impact of South Korea’s national insurance coverage (NIC) expansion and the addition of genetic counselors on *BRCA1/2* mutation testing rates in breast cancer patients. Materials and Methods: A retrospective review was conducted at the Samsung Medical Center (SMC), dividing patients into three groups: pre-NIC expansion, post-NIC expansion, and post-extra genetic counselor involvement. The number of *BRCA1/2* tests performed and the detection rates among newly diagnosed and follow-up patients, particularly focusing on triple-negative breast cancer (TNBC) cases, were analyzed. Results: Post-NIC expansion, there was a significant increase in *BRCA1/2* testing rates, with a gradual rise in detection rates while maintaining statistical significance. TNBC patients under 60 experienced substantial increases in testing rates. The number of follow-up patients recalled for testing also rose significantly after the extra genetic counselor involvement. Additionally, NIC expansion increased insurance coverage for TNBC patients, enhancing accessibility to testing. Conclusion: The study highlights the positive impact of NIC expansion and genetic counselor involvement on *BRCA1/2* mutation testing rates and subsequent patient management. Addressing financial barriers to testing and incorporating genetic counseling significantly improve patient outcomes. This model provides a potential strategy for enhancing early detection and personalized treatment for breast cancer patients with *BRCA1/2* mutations, contributing to global cancer management efforts.

## 1. Introduction

Breast cancer remains one of the most prevalent and concerning diseases affecting women worldwide [1,2]. The risk of breast cancer in the general population is about 12%; however, when a *BRCA1/2* mutation is present, the risk increases dramatically. *BRCA1* mutations appear in 50–80% and *BRCA2* mutations in 40–70% of breast cancer risks. *BRCA1* and *BRCA2* mutations account for about 40–50% of hereditary breast cancers and about 5–10% of all breast cancers. Even though many are not yet diagnosed with breast cancer, approximately 46% of BRCA1 mutation carriers and 52% of *BRCA2* mutation carriers are diagnosed with breast cancer at some point in their lifetime. Thus, the early detection of *BRCA1/2* mutations comes to the forefront nowadays [3,4,5,6]. However, amidst the growing awareness and demand for *BRCA1/2* mutation testing, the issue of its high cost has emerged as a substantial barrier, limiting access to this crucial tool for many individuals. Therefore, the public insurance coverage of *BRCA1/2* mutation testing becomes an important issue.

In a previous study, the high prevalence of *BRCA1/2* mutations in Korean TNBC patients diagnosed under the age of 60 was analyzed, and this led to the Korean NIC expansion in July 2020 [7,8,9]. Along with the insurance coverage issue, genetic counseling before and after a genetic test is also important. To explain and recommend *BRCA1/2* mutation testing to patients who meet the criteria, as well as to provide quality counseling after *BRCA1/2* mutation testing, the need for genetic counselors is inevitable [10]. Since genetic counseling is an important but time-consuming process, it is difficult to provide counseling to a sufficient number of patients, especially when a limited number of counselors are present.

The purpose of this study is to evaluate the effect of the Korean NIC expansion and the additional recruitment of genetic counselors on the increase in *BRCA1/2* mutation tests and *BRCA1/2* mutation rates.

## 2. Materials and Methods

### 2.1. Patient Selection

This is a single-institution retrospective review. All patients who received breast cancer surgery from August 2019 to December 2021 at Samsung Medical Center (SMC) were divided into three groups according to the time marks. Group 1 was patients who underwent primary breast cancer surgery at our center before the Korean NIC expansion (August 2019–June 2020), group 2 consisted of patients after the Korean NIC expansion but before additional genetic counselor involvement (July 2020–November 2020), and group 3 consisted of patients after additional genetic counselor involvement (December 2020–December 2021). Among the total number of patients who received primary breast cancer surgery in each time period, patients who underwent *BRCA1/2* mutation tests were selected and assessed according to the Korean NIC criteria on the *BRCA1/2* mutation test. Moreover, follow-up patients who were recalled for *BRCA1/2* mutation tests were selected during two different time periods: after the Korean NIC expansion (July 2020–November 2020) and after additional genetic counselor involvement (December 2020–December 2021). We also went through the numbers of new TNBC patients tested for the *BRCA1/2* mutation who were within and out of insurance coverage before and after the Korean NIC expansion.

### 2.2. Rationale for Patient Selection and Criteria

In a previous clinical trial at our center, we analyzed the higher prevalence of *BRCA1/2* mutations in Korean TNBC patients diagnosed under the age of 60, and this led to the Korean NIC expansion [7]. After the Korean NIC expansion, the 3 major changes in *BRCA1/2* mutation test criteria became the following (Table 1) [11,12]:

Breast cancer diagnosed ≤40 years old;

Breast cancer patient with family history of breast cancer, ovarian cancer, metastatic prostate cancer, or pancreatic cancer (within 3rd degree);

TNBC diagnosed ≤60 years old.

**Table 1 cancers-16-01865-t001:** The effect of national insurance coverage changes on *BRCA1/2* gene mutation testing in Korea.

April 2012–July 2020	After July 2020
BC diagnosed <40 years old	BC diagnosed ≤40 years old
BC patients with FHx of BC or OC (within 2nd degree)	BC patients with FHx of BC, OC, metastatic prostate cancer, or pancreatic cancer (within 3rd degree)
Personal history of BC and OC	Personal history of BC and/or OC
Male BC	Male BC
Bilateral BC	Bilateral BC
	TNBC diagnosed ≤60 years old
Serous OC patient	Serous OC patient

BC, breast cancer; FHx, family history; OC, ovarian cancer; TNBC, triple-negative breast cancer.

### 2.3. Group Selection and Comparison

To obtain statistical significance in the comparison of each of the group, which has different patient number pools due to the different time periods set, the proportion of patients instead of the exact number of patients was compared. Only newly diagnosed breast cancer patients were selected to analyze the increase in the *BRCA1/2* mutation testing number in each group, and old patients undergoing regular OPD follow-ups who met the new insurance criteria were selected to analyze the recall rate after the NIC expansion as well as after additional genetic counselor involvement. Since the age expansion in TNBC patients was one of the major changes in the Korean NIC expansion, the number of TNBC patients was counted separately to see how effective the NIC expansion was on *BRCA1/2* mutation testing in new TNBC patients.

### 2.4. Statistics

The *BRCA1/2* mutation testing numbers in each group were assessed with Pearson’s chi-squared test. All statistical analyses were performed using R Statistical Software (version 3.6.3; Foundation for Statistical Computing, Vienna, Austria). Statistical significance was accepted for *p*-values of <0.05.

### 2.5. Ethics

This study adhered to the ethical tenets of the Declaration of Helsinki and was approved by the Institutional Review Board (IRB) of SMC (IRB number: 2023-01-103-001). The need for informed consent was waived because of the retrospective nature of this study.

## 3. Results

A total of 7299 patients received breast cancer surgery from August 2019 to December 2021 in the SMC, and they were divided into three groups according to the time marks. Group 1 consists of 2539 patients who underwent primary breast cancer surgery at our center before the Korean NIC expansion (August 2019–June 2020), group 2 consists of 1164 patients after the NIC expansion but before additional genetic counselor involvement (July 2020–November 2020), and group 3 consists of 3596 patients after additional genetic counselor involvement (December 2020–December 2021).

### 3.1. Major Changes in BRCA1/2 Mutation Testing Numbers in Newly Diagnosed Breast Cancer Patients after the Korean NIC Expansion and Additional Genetic Counselor Involvement

The total number of patients who underwent *BRCA1/2* mutation tests at the SMC dramatically increased during 2020–2021, which reflects the time period after the Korean NIC expansion (Figure 1). Before the Korean NIC expansion, 32.8% of new patients who underwent primary breast cancer surgery were tested for the *BRCA1/2* mutation. Then, 37.8% were tested after the Korean NIC expansion, and 42% were tested after additional genetic counselor involvement, respectively. The increase in the number of *BRCA1/2* mutation tests in each group was statistically significant (*p*-value: 0.003 (group 1 vs. 2), 0.011 (group 2 vs. 3), <0.001 (group 1, 2 vs. 3). Even though the *BRCA1/2* mutation testing numbers have increased, the *BRCA1/2* mutation detection rate has remained similar (9.6%, 9.3%, and 9.5% in each group; *p*-value: 0.92 (group 1 vs. 2), 1.00 (group 2 vs. 3), 1.00 (group 1, 2 vs. 3)) (Table 2). Criteria-wise, breast cancer patients diagnosed under the age of 40, bilateral breast cancer patients, and patients with a personal history of ovarian/pancreatic cancer did not show a statistically significant increase in the number of *BRCA1/2* mutation tests. However, there was a significant increase in the *BRCA1/2* mutation testing numbers among TNBC patients diagnosed under the age of 60: 0.8% before the Korean NIC expansion, 5.2% after the Korean NIC expansion, and 15.3% after additional genetic counselor involvement (Table 2). To examine statistical differences across the three groups, Pearson’s chi-squared test followed by a pairwise comparison was conducted, and the results showed statistically significant differences across the three groups for TNBC patients aged 60 or younger (*p* < 0.001). According to our data on insurance coverage in TNBC patients tested for the *BRCA1/2* mutation since 2016, there has been a dramatic reversal in the number of patients who were covered with insurance or not at the time of the Korean NIC expansion. Interestingly, as patients with a family history (within the third degree) of metastatic prostatic or pancreatic cancer were included in the expanded insurance criteria, the number of *BRCA1/2* mutation tests carried out on these patients increased dramatically from 0.4% before the Korean NIC expansion to 8.9% after the Korean NIC expansion and remained about the same (7.4%) after additional genetic counselor involvement.

### 3.2. Follow-Up Patients Recalled for BRCA1/2 Mutation Testing

Among the total of 2536 patients who underwent *BRCA1/2* mutation tests, which includes both new and old patients, 584 follow-up patients were recalled for *BRCA1/2* mutation testing under new insurance criteria, with a mean *BRCA1/2* mutation detection rate of 7.7%. Among all of the recalled patients, 538 (92.8%) of follow-up patients were recalled and tested for the *BRCA1/2* mutation after additional genetic counselor involvement (Table 3).

### 3.3. Effect of Insurance Coverage Expansion on BRCA1/2 Mutation Testing in New TNBC Patients

Before the Korean NIC expansion, when only TNBC patients under the age of 40 were included in the *BRCA1/2* mutation testing criteria, only 34.1% of TNBC patients were within the insurance coverage range and could benefit from *BRCA1/2* mutation tests. However, there was a dramatic increase in this patient percentage to 82.1% being covered by insurance after the Korean NIC criteria, which included TNBC patients between the ages of 40 and 60 (Table 4).

## 4. Discussion

The NIC expansion had a significant role in increasing the total number of patients under the criteria to be tested for the *BRCA1/2* mutation while maintaining a similar detection rate (about 9%, Table 2), which clearly shows that the proper tests were being carried out under the proper criteria [11,12]. This increase was even more significant in TNBC patients and in breast cancer patients with a family history of metastatic or pancreatic cancer [12,13]. Moreover, the additional recruitment of genetic counselors led to a dramatic increase in recalling follow-up patients to be tested for the *BRCA1/2* mutation.

Our results show that the Korean NIC expansion and additional genetic counselor involvement together brought about a significant increase in the number of effective *BRCA1/2* mutation tests being provided to newly diagnosed breast cancer patients. The observed gradual increase in the total number of *BRCA1/2* mutation tests, alongside a proportional rise in the testing frequency, while maintaining a stable *BRCA1/2* mutation detection rate, serves as evidence that appropriate *BRCA1/2* mutation tests are being conducted for eligible patients according to established criteria. It is noteworthy that the presence or absence of NIC for the *BRCA1/2* mutation tests varies globally. Taiwan lacks NIC for genetic tests (including for the *BRCA1/2* mutation). This shared characteristic highlights the financial burden on patients. Therefore, the strength of gene mutation test recommendations remains low, and *BRCA1/2* mutation tests are not recommended to all patients meeting the NCCN criteria but only to those who can afford them financially. An absence of NIC is also observed in Vietnam and Singapore, where patients may face similar financial challenges [14]. Conversely, Japan, akin to Korea, provides NIC for *BRCA1/2* mutation tests. This shared characteristic sets Japan and Korea apart from the other countries in this comparison. Even though the availability and structure of NIC for *BRCA1/2* mutation tests may vary among these nations, professional genetic counseling and the referral of *BRCA1/2*-unaffected carriers and *BRCA1/2*-positive patients to specialized departments are available in all of the countries mentioned above. This diverse landscape of insurance coverage and genetic counseling practices emphasizes the need to consider the global context. The adoption of genetic tests and counseling for cancer patients still remains low in Asia due to several difficulties: financial reasons, lack of time and resources, and limited access to genetic counselors [15]. If NIC expansion lowered the financial barriers to genetic testing for cancer patients and more professional genetic counselors were available for proper genetic counseling in Asia, more patients would be able to benefit from it.

In terms of genetic counseling, the framework of counseling itself has not changed dramatically. However, since preliminary counseling (before the patient visits a doctor’s office) was added to the counseling flow, triple checks and counseling have become possible with the additional recruitment of genetic counselors. Also, more thorough pre- and post-genetic counseling is now possible by reinforcing the manpower of specialists (Figure 2). It is also clear from the results that the Korean NIC expansion and genetic counseling’s role together have important effects not only on new patients but also on follow-up patients who can be recalled for testing under the new insurance criteria with the support of professional counseling [16,17]. A great increase in the recall of follow-up patients at our center was possible after recruiting additional genetic counselors. The recall of follow-up patients for *BRCA1/2* mutation tests is meaningful since about 8% of newly tested patients could be detected as *BRCA*-positive, which would lead to prophylactic treatments and an increase in the overall survival rate [18]. The international comparison of insurance coverage and genetic counseling practices further highlights the importance of addressing the financial aspect of *BRCA1/2* mutation testing, as access to such tests is a critical factor in early cancer detection and proper management. The differences in NIC and the availability of counseling services across countries underline the need for a global perspective in the management of breast cancer patients. Our positive results regarding the Korean NIC expansion could serve as a potential model for other nations aiming to improve early detection and personalized treatment for breast cancer patients with *BRCA1/2* mutations.

The proper detection of the *BRCA1/2* mutation status of patients helps clinicians to make quick but informed and timely decisions for patients [19,20], which gradually leads to appropriate treatments. Also, the identification of the *BRCA1/2* mutation status can not only inform family members of *BRCA1/2* mutation carriers about their potential cancer risks but also help identify those who could benefit from active surveillance or risk reduction strategies, such as risk-reducing surgeries for breasts and ovaries [5,21,22,23]. For the early detection of the *BRCA1/2* mutation status to be possible, financial aspects should be addressed, foremost since a *BRCA1/2* mutation test is a costly process. If the Korean NIC could be expanded even more within reasonable standards, more patients would benefit from it. The effectiveness of surveillance in detecting breast cancer in *BRCA1/2* mutation carriers was evaluated in 2017. The study included 2482 *BRCA1/2* mutation carriers and found that enhanced surveillance with annual breast MRI scans and mammography resulted in a higher likelihood of detecting early-stage breast cancers. Specifically, the study showed that surveillance with breast MRI detected 56% of breast cancers in *BRCA1* mutation carriers and 73% in *BRCA2* mutation carriers. Additionally, mammography detected 33% of breast cancers in *BRCA1* mutation carriers and 43% in *BRCA2* mutation carriers [24].

*BRCA1/2* mutation testing is of significant importance to breast cancer patients for several reasons, as it can impact their treatment decisions and overall management [17,18]. Gentile et al. reported that the rate of preoperative genetic diagnosis turned out to be only 21.8% in a large series of *BRCA1/2* mutation carriers, with a great impact on the subsequent management of these patients [25]. The ability to ascertain the *BRCA1/2* mutation status prior to surgery allows for informed genetic counseling and discussion of surgical options. This further underscores the importance of *BRCA1/2* mutation testing in guiding clinical decision-making processes [26]. Hartmann et al. investigated the risk of subsequent breast cancer in *BRCA1/2* mutation carriers who underwent a risk-reducing mastectomy [27]. The study found that among the *BRCA1* mutation carriers, a risk-reducing mastectomy was associated with a 90% reduction in the risk of developing subsequent breast cancer (incidental cancer) during a median follow-up of 3.6 years [28]. For *BRCA2* mutation carriers, a risk-reducing mastectomy was associated with an 86% reduction in the risk of developing subsequent breast cancer during the same follow-up period [29]. However, there remains ongoing debate regarding the optimal surgical management for *BRCA1/2* mutation carriers, as evidence suggests that different surgical approaches (breast conserving surgery vs. bilateral mastectomy) may not significantly impact the oncological outcomes [25]. Therefore, the ability to ascertain the *BRCA1/2* mutation status before surgery and engage in appropriate genetic counseling for surgical decision making would indeed confer significant advantages in patient management. Furthermore, recent findings by Apostolova et al. underscore the impact of preoperative genetic testing on surgical decision making. Their conclusion that genetic testing results delivered prior to index breast surgery increase the uptake of bilateral risk-reducing mastectomy (RRM) in affected *BRCA1/2* and *PALB2* carriers [30] aligns closely with our discussion on the importance of ascertaining the BRCA1/2 mutation status before surgery. This highlights the critical role of genetic testing in optimizing surgical decision making and underscores the need for efforts to make genetic testing more mainstream in clinical practice. In terms of treatment, the OlympiA trial investigated the effectiveness of adding olaparib, a targeted therapy, to standard therapy in breast cancer patients with *BRCA1/2* mutations. The trial results showed that the addition of olaparib reduced the risk of invasive disease recurrence or death by 42% in patients with high-risk, early-stage breast cancer and *BRCA1/2* mutations. According to the study, the invasive disease-free survival rate was 87.5% in the olaparib group compared to 77.1% in the placebo group after a 3-year follow-up. Also, the absolute improvement in invasive disease-free survival at three years with olaparib was 9.4%. These results suggest that olaparib benefited both groups of *BRCA1/2* mutation carriers [31].

These findings demonstrate the importance of *BRCA1/2* mutation testing in breast cancer patients, as identifying *BRCA1/2* mutations can lead to personalized treatment decisions, such as the use of targeted therapies like olaparib, which have shown significant benefits in reducing the risk of recurrence and improving survival rates in patients with *BRCA1/2* mutations.

## 5. Conclusions

In conclusion, the expansion of the Korean NIC and the concurrent recruitment of additional genetic counselors have had a significant and beneficial effect on the number of *BRCA1/2* mutation tests conducted at the SMC. The results clearly indicate a gradual increase in the number of tests performed, all while maintaining a sufficient positive detection rate. This outcome underscores the positive impact of these measures, eventually leading to more tailored treatments for affected patients.

## Figures and Tables

**Figure 1 cancers-16-01865-f001:**
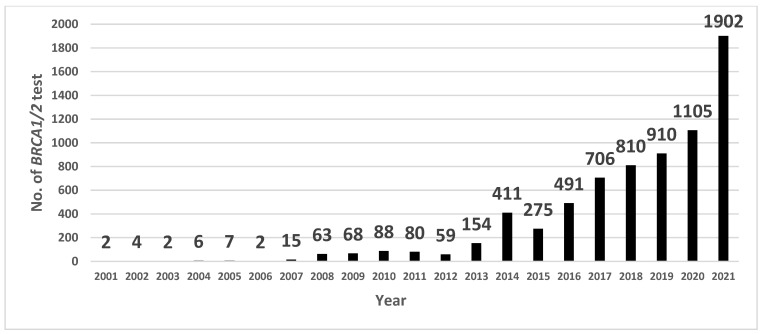
Number of *BRCA1/2* gene mutation tests on breast cancer patients at Samsung Medical Center (SMC).

**Figure 2 cancers-16-01865-f002:**
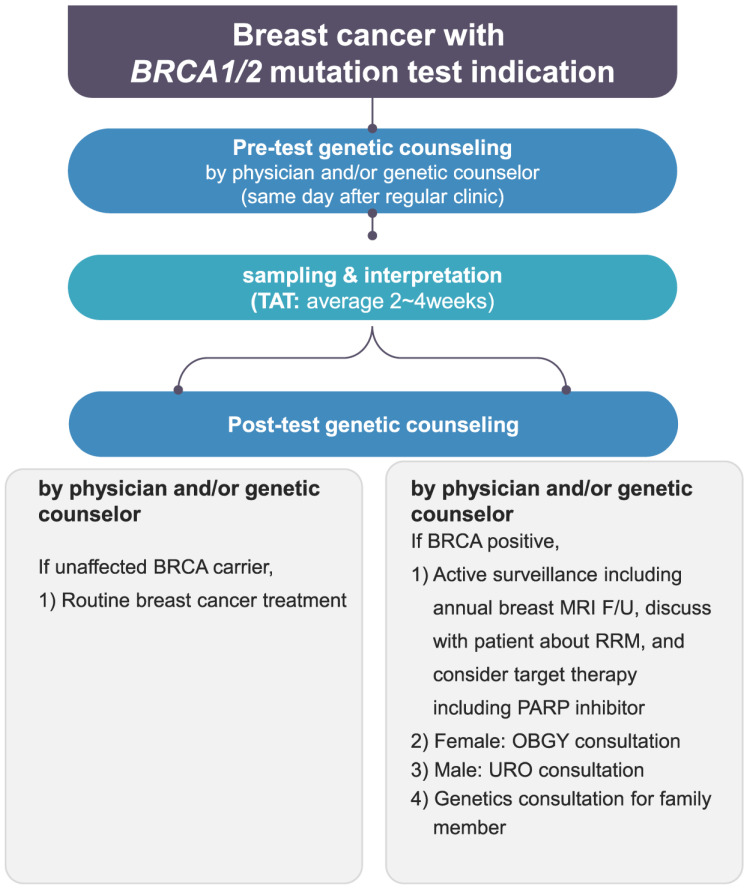
Genetic counseling flowchart at the Breast Center of Samsung Medical Center. TAT, turn around time; MRI, magnetic resonance imaging; F/U, follow-up; RRM, risk-reducing mastectomy; URO, urology; OBGY, obstetrics and gynecology.

**Table 2 cancers-16-01865-t002:** Changes in *BRCA1/2* gene mutation testing numbers before and after NIC expansion and after genetic counselor involvement (among new patients who underwent primary breast cancer surgery).

	Before NIC Expansion	After NIC Expansion	After Additional Genetic Counselor Involvement	*p*-Value			*p*-Value for Multivariate
	Group 1 (August 2019–June 2020)	Group 2 (July 2020–November 2020)	Group 3 (December 2020–December 2021)	a vs. b	b vs. c	a, b vs. c	
Total number of breast cancer surgeries held at SMC	2539	1164	3596				
Number of *BRCA1/2* tests among all primary breast cancer surgeries	833 (32.8)	440 (37.8)	1512 (42.0)	0.003	0.011	<0.001	
*BRCA1/2* results:							
*BRCA1/2* PV/LPV, number (%)	80 (9.6)	41 (9.3)	144 (9.5)	0.920	1.000	1.000	0.986
*BRCA1/2* VUS, number (%)	52 (6.2)	33 (7.5)	75 (5.0)	0.410	0.044	0.06	0.100
*BRCA1/2* Negative, number (%)	704 (84.5)	375 (85.2)	1296 (85.7)	0.806	0.817	0.486	0.734
Age at diagnosis ≤40 years old (%)	355 (42.6)	173 (39.3)	595 (39.4)	0.282	1.000	0.261	0.273
TNBC ≤60 years old	7 (0.8) _b,c_	23 (5.2) _a,c_	232 (15.3) _a,b_	<0.001	<0.001	<0.001	<0.001
Bilateral BC, *n* (%)	136 (16.3)	65 (14.8)	208 (13.8)	0.518	0.585	0.133	0.242
Personal history of OC, *n* (%)	2 (0.2)	2 (0.5)	5 (0.3)	0.612	0.659	1.000	0.812
Personal history of PC, *n* (%)	2 (0.2)	0 (0)	0 (0)	0.547	1.000	0.209	0.096
Family history of BC and/or OC, *n* (%)	428 (51.4)	216 (49.1)	619 (40.9)	0.444	0.003	<0.001	<0.001
Family history of metastatic prostate canceror pancreatic cancer, *n* (%)	3 (0.4)	39 (8.9)	112 (7.4)	<0.001	0.312	<0.001	<0.001

NIC, national insurance coverage; SMC, Samsung Medical Center; PV, pathogenic variant; LPV, likely pathogenic variant; VUS, variants of uncertain significance; TNBC, triple-negative breast cancer; BC, breast cancer; OC, ovarian cancer; PC, pancreatic cancer. _a_ = group 1; _b_ = group 2; _c_ = group 3.

**Table 3 cancers-16-01865-t003:** Follow-up patients recalled for *BRCA1/2* gene mutation testing after NIC expansion.

	After NIC Expansion (July 2020–November 2020)	After Additional Genetic Counselor Involvement (December 2020–December 2021)	*p*-Value
Total number of *BRCA1/2* tests for new and old patients	486	2050	
Number of *BRCA1/2* tests among pre-NIC expansion patients satisfying new criteria (%)	46 (9.5)	538 (26.2)	<0.001
*BRCA1/2* results:			
*BRCA1/2* PV/LPV, number (%)	5 (10.9)	40 (7.4)	0.386
*BRCA1/2* VUS, number (%)	2 (4.3)	40 (7.4)	0.764
*BRCA1/2* Negative, number (%)	39 (8.0)	460 (85.5)	0.829

NIC, national insurance coverage; PV, pathogenic variant; LPV, likely pathogenic variant; VUS, variants of uncertain significance.

**Table 4 cancers-16-01865-t004:** Insurance coverage in TNBC for new patients.

	January 2016–July 2020	July 2020–December 2021	*p*-Value
Within insurance coverage	483 (34.1)	445 (82.1)	<0.0001
Out of insurance coverage	935 (65.9)	97 (17.9)	
Total	1418	542	

TNBC, triple-negative breast cancer.

## Data Availability

The original contributions presented in the study are included in the article, and further inquiries can be directed to the corresponding authors.

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
