# Peer review of "The Effects of National Insurance Coverage Expansion and Genetic Counseling’s Role on *BRCA1/2* Mutation Tests in Breast Cancer Patients"

_cancers, 2024, doi:10.3390/cancers16101865_

Round 1
Reviewer 1 Report
Comments and Suggestions for Authors
The study entitled "The Effects of National Insurance Coverage Expansion and Genetic Counselor’s Role on BRCA1/2 Mutation Test in Breast Cancer Patients" performed by Jang et al. shows the impact of the Korean National Insurance Coverage (NIC) expansion and the integration of additional genetic counseling on BRCA1/2 mutation testing in breast cancer patients.
The findings demonstrate a significant increase in the number of BRCA1/2 tests conducted post-NIC expansion and following the involvement of more genetic counselors, without altering the mutation detection rate. Testing among TNBC patients and those with a family history of related cancers saw a considerable rise.
I have some comments:
- The Introduction paragraph is too brief and needs expansion and more context;
- Surely the issue of pre-operative genetic diagnosis is very critical. It is reported to be only 21.8% in some large series of BRCA-mutation carriers (PMID: 35534308) with a great impact on subsequent management of these patients. This issue needs to be addressed. On the other hand, the choice of breast surgery (breast conserving surgery vs bilateral mastectomy) does not seem to influence the oncological outcomes. Please cite the suggested reference and elaborate this key topic to improve the quality of your manuscript;
- Why you did not develop multivariate analysis between groups (group 1 vs group 2; group 2 vs group 3, and group 1,2 vs group 3) since you have statistically significant characteristics at univariate analysis. Please integrate with a multivariate analysis to improve your results.
Reviewer 2 Report
Comments and Suggestions for Authors
The authors investigated the effects of National Insurance Coverage expansion and genetic counselors on BRCA1/2 mutation testing rates in breast cancer patients in South Korea. The results indicated s a positive impact of National Insurance Coverage expansion and genetic counselor involvement on BRCA1/2 mutation testing rates. The study selected the patients receiving breast cancer surgery from August, 2019 to December, 2021 in Samsung Medical Center, can the authors start the patient’s recruitment and analysis earlier than 2019, and expand the study to 2023? Since it was concluded that National Insurance Coverage expansion and genetic counselors can improve patient outcomes, the impact of National Insurance Coverage expansion, genetic counselors, and testing rates on the prognosis of patients, such as survival and recurrence, should be investigated and deeply discussed.
Reviewer 3 Report
Comments and Suggestions for Authors
1. Lines 119-130 should be moved to the Methods section.
2. Why do the authors provide data on ovarian and prostate cancer? The article is about breast cancer, the rest of the information is unnecessary.
3. Does Figure 1 show the data from all tests for the BRCA1/2 mutation? or just for breast cancer? If that’s all, then you should leave only the statistics on breast cancer.
4. Is there a typo in the table in 39 (8.0) in brackets? The column does not converge based on the sum %.
5. In table 4, columns 2020.07-2021.12 and 2016.01-2020.07 must be swapped.
6. It turns out that the % detection of the BRCA1/2 mutation did not change in the three groups (according to Table 1). The improvement was achieved only due to a general increase in the number of tests, did I understand correctly? In this regard, it is interesting to see how survival rates among patients have changed.
Round 2
Reviewer 1 Report
Comments and Suggestions for Authors
Accept in present form
Reviewer 2 Report
Comments and Suggestions for Authors
It is well revised.